# Optimising the adult HIV testing services screening tool to predict positivity yield in Zimbabwe, 2022

**Hamufare Dumisani Mugauri**[1,2¤]*, **Joconiah Chirenda**[1], **Kudakwashe Takarinda**[3], **Owen Mugurungi**[2], **Getrude Ncube**[2], **Ishmael Chikondowa**[2], **Patrick Mantiziba**[4], **Blessing Mushangwe**[5], **Mufuta Tshimanga**[1]

**1** Department of Primary Health-care Sciences, The University of Zimbabwe, Harare, Zimbabwe, **2** Ministry of Health and Child Care, AIDS and TB Unit, Harare, Zimbabwe, **3** Organisation for Public Health Interventions and Development (OPHID), Harare, Zimbabwe, **4** Clinton Health Access Initiative (CHAI), Harare, Zimbabwe, **5** Zimbabwe Technical Assistance, Training and Education Center for Health (Zim-TTECH), Harare, Zimbabwe

¤ Current address: Faculty of Medicine and Health Sciences, Department of Primary Health-care Sciences, Harare, Zimbabwe
* dumiwaboka@gmail.com

**Data Availability Statement:** The dataset used in this study has been provided in S1 Data.

**Funding:** The author(s) received no specific funding for this work.

## Abstract

HIV positivity yield declined against increasing testing volumes in Zimbabwe, from 20% (1.65 million tests) in 2011 to 6% (3 million tests) in 2018. A screening tool was introduced to aid testers to identify clients likely to obtain a positive diagnosis of HIV. Consequently, testing volumes declined to 2.3 million in 2019 but positivity declined to 5% prompting the evaluation and validation of the tool to improve its precision in predicting positivity yield. A cross-sectional study was conducted. Sixty-four sites were randomly selected where all reporting clients (18+ years) were screened and tested for HIV. Participant responses and test outcomes were documented and uploaded to excel. Multivariable analysis was used to determine the performance of individual, combination questions and screening criteria to achieve >/= 90% sensitivity for a new screening tool. We evaluated 13 questions among 7,825 participants and obtained 95.7% overall sensitivity, ranging from 3.9% [(95%CI:2.5,5.9) sharing sharp objects] to 86.8% [(95%CI:83.8,89.5) self-perception of risk] for individual questions. A 5-question tool was developed and validated among 2,116 participants. The best combination (self-perception of risk, partner tested positive, history of ill health, last tested >/= 3months and symptoms of an STI) scored 94.1% (95%CI:89.4,97.1) sensitivity, 18% reduction in testing volumes and 11 Number Needed to Test (NNT). A screening in criteria that combine previously testing >/= 3 months with a yes to any of the 4 remaining questions was analysed and sensitivity ranged from 89.9% (95%CI:84.4,94.0) for last tested >/= 3months and sexual partner positive, to 93.5% (95%CI:88.7,96.7) for last tested >/= 3months and self-perceived risk We successfully developed, evaluated and validated an HIV screening tool. High sensitivity and the fifth reduction in testing volume were acceptable attributes to enhance testing efficiency and effective limited resource utilisation. Screened out clients will be identified through frequent screening and self-testing options.

**Competing interests:** The authors have declared that no competing interests exist.

## Introduction

Zimbabwe's HIV programmatic data indicate that Adult positivity yield has been on a descending trajectory, from 20% in 2011 to 6% in 2018 and 2020, a situation that is attributable to exhausted low hanging fruits among other factors, making it more difficult to identify the remaining individuals living with HIV [1]. However, HIV remains firmly established at 12.9% adult prevalence and an incidence of 0.38% (0.54 per cent among women and 0.20 per cent among men) translating to approximately 31,000 new infections per year [2].

Testing for HIV is the critical entry point to results tailored interventions for prevention and treatment. In 2016, Zimbabwe launched a multi-pronged approach to HIV testing that includes Provider Initiated Testing and Counselling (PITC) Client-Initiated Testing and Counselling (CITC) Index contact tracing and testing coined Index testing (ICT) and HIV Self-testing (HIVST) as part of a consortium of measures designed to enhance identification of people living with HIV as encapsulated in the Antiretroviral Therapy (ART) Guidelines, 2016 [3].

Further, the HIV testing approach was shifted from testing for coverage to targeted testing as recommended by WHO to enhance the identification of remaining people living with HIV and expedite epidemic control [4, 5]. Key to the targeted testing stratagem is the use of Screening tools to aid health workers in determining eligibility for testing and only test clients with a high probability of testing HIV positive [6, 7].

The concept of screening for eligibility for an HIV test was introduced in 2019, but was deficient of a clearly defined criteria for screening clients in and out of testing, neither were the questions assessed for their performance. This therefore resulted in this screening approach reducing the testing volumes from 3 million in 2018, to 2.3 million in 2019. Consequently, positivity yields also declined from 6% in 2018 to 5% in 2019, raising concern that the screening process was screening out clients who would have otherwise tested HIV positive. (**Fig 1**) The questions' performance was unknown.

Evaluation of screening questions is critical to ascertain its performance attributes that include sensitivity and specificity, predictive value positive (PVP) and negative (PVN) and Number Needed to Test (NNT) as critical attributes that validate its usefulness and value addition [8–10]. Ideally, a good screening tool should be at least 90% sensitive and screen out <10% of potential positive testers who would be captured through other innovations or re-screening at 3 months intervals.

We, therefore sort to evaluate and validate the adult HTS screening tool to develop a screening tool with at least 90% sensitivity and <10% screened out potential positive testers, and specifically to;

- Revise the existing screening tool and add more questions for evaluation

- Identify the questions that have the highest sensitivity to develop a new screening tool of >90% sensitivity

- Validate the developed screening tool and measure its attributes (sensitivity, specificity, PVP & PVN, proportion of positive testers screened out and NNT)

- Determine the screening in and screening out criteria

- Recommend the developed screening tool for adoption and utilisation by the Ministry of Health and Child Care, Zimbabwe

## Materials and methods

### Study design

We conducted a cross-sectional study, with an analytical component.

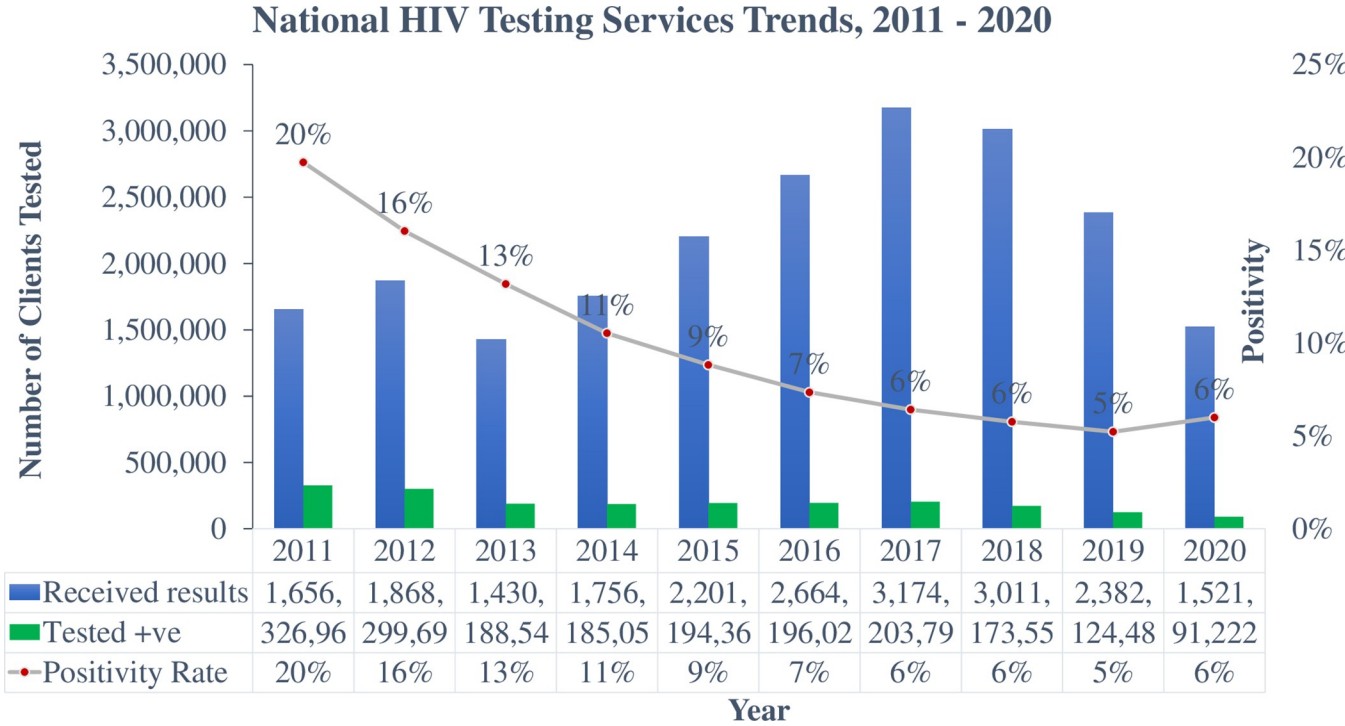

**Fig 1. Zimbabwe national HIV testing services trends, 2011–2020 (source: HIV prevention annual program performance report, 2020).**

## Setting

**General setting.** Zimbabwe is a landlocked, low-income country in Southern Africa which is located between Botswana, South Africa, Mozambique and Zambia with an estimated population of 16 million and a human development index of 0.516, ranked number 154 globally out of 189 countries in 2016 [11]. The country is divided into two urban provinces, eight rural provinces and 62 districts. The capital city is Harare and other major cities include Bulawayo, Gweru, Kadoma, Kwekwe, Masvingo and Mutare [12].

**Zimbabwe national HIV programme.** The AIDS and TB Programme (ATP) coordinates the development of HIV/AIDS health policies and the setting up of national standards and guidelines as part of the national response to HIV in Zimbabwe. Four sub-units under ATP, namely; HIV Prevention, Care and Treatment, Prevention of Mother to Child Transmission (PMTCT) and Monitoring and Evaluation (M & E) are mandated to ensure seamless yet specialised programming to ensure adequate response to the pandemic [13].

All clients who report at the public health facilities are offered HIV testing services after being screened for eligibility, according to existing Job aides and Operational Service Delivery Manual (OSDM) [14]. Provider initiated testing and counselling (PITC) is practised at the facility and in the community, whereby the health worker makes the initiative to offer HIV testing services to eligible clients regardless of the purpose of visit. Clients may also demand the service (Client-Initiated Testing and Counselling—CITC) [15]. HIV screening results are not routinely documented, the process only aides the service provider to determine if the client can be tested during that visitor to be advised to report back at a later date, according to their risk profile.

Outpatients (OPD), Family and Child Health (FCH) departments, as well as Opportunistic Infections clinics (OIC), are the common entry points for HTS. Admitted clients may also be

tested within the wards. Community health workers also offer HIV testing within the community through outreach initiatives and where indicated, door testing programs.

All HIV tested clients are documented with a unique identification number in a paper-based register whilst test results are issued using client intake forms and may also be documented in the client's treatment booklet. The paper-based register and client intake form are the source documents for electronic entry into DHIS2 [16]. The country is however rolling out electronic Health Records (eHR) aimed at migrating to electronic collection of primary data using the entry first approach. Paper-based registers are in the process of being decommissioned in areas that have fully migrated to eHR.

**Study site.** The study was conducted in 64 randomly selected sites across the 10 provinces of the country, to provide a mix of urban as well as rural provinces. The study sites were HIV testing entry points at the facility and community level. At the facility level, the entry points utilized for the routine provision of HIV testing services were the same used during the study to provide a natural environment that will reflect routine service provision circumstances and minimize bias.

## Client population

All adult clients (18+ years) who reported to the participating health facilities during the evaluation and validation phases were screened and offered an HIV test, except for known HIV positives.

## Sampling

Seven districts were randomly selected per province, from which 1 facility was randomly selected for inclusion in the study. A total of 70 health facilities were therefore identified and included in the study.

The sampling criteria took into consideration the need to include facility and community entry points, high and low volume facilities to obtain a wide variety of responses.

## Data variables, sources of data and data collection

Data were collected from 1 to 30 April 2021 for the initial phase (evaluation) using a structured data collection form, which was the developed draft screening tool (**S1 Text**). The questions were extrapolated from commonly used risk profiling questions both locally and regionally, resulting in a total of 13 questions being evaluated. These were printed and used as data collection tools where health workers documented the response to questions and also the HIV test results.

The second phase of data collection (validation) was conducted from 4 to 15 October 2021. An evaluation of the 13 questions was conducted to assess their performance (**Table 1**). Five questions scored a predetermined sensitivity of >/ = 40% and were included in the validation process (**S2 Text**). The questions were reviewed and adjusted to improve precision as follows;

- The previous HIV testing period was adjusted from the initial below one year and above 1 year to include the following categories; 0-3months, 3-6months, 6months-1 year, >12 months and never

- Experiencing ill health within the previous 1 year was adjusted to 3 months and to include presumptive symptoms of an STI. This was further adjusted in the SOP to guide the data collectors

**Table 1. Sensitivity and specificity for preliminary screening questions, Zimbabwe, 2021.** (N = 7,825) #.

| Variable | HIV Positive | Negative | Positivity Yield (%) | Sensitivity % (95% CI) [&] | Specificity % (95% CI) |
|---|---|---|---|---|---|
| Total | 584 | | 7.52 | | |
| When was the last time you were tested for HIV? | | | | | |
| ○ >/ = 1 year | 370 | 3,348 | 9.9 | 63.4 (59.3–67.3) | 53.4 (52.2–54.5)[^] |
| ○ < 1 year | 214 | 3,830 | 5.3 | | |
| Do you consider yourself to be at risk of HIV infection? [€] | | | | | |
| ○ Yes | 507 | 5,304 | 8.7 | 86.8 (83.8–89.5) | 26.1 (25.1–27.1)[^] |
| ○ No | 77 | 1,874 | 4 | | |
| In the past 1 year, have you had more than 1 sexual partner concurrently? | | | | | |
| ○ Yes | 105 | 582 | 15 | 18 (14.9–21.3) | 91.9 (91.2–92.5) |
| ○ No | 479 | 6,596 | 6.7 | | |
| Have you experienced poor health in the past 1 year? | | | | | |
| ○ Yes | 246 | 1,514 | 13.7 | 42.1 (38.1–46.2) | 78.9 (77.9–79.8)[^] |
| ○ No | 338 | 5,664 | 7.2 | | |
| In the past 1 year, have you experienced recurrent skin rashes? | | | | | |
| ○ Yes | 41 | 198 | 16.7 | 7 (5.1–9.4) | 97.2 (96.8–97.6) |
| ○ No | 543 | 6,980 | 7.2 | | |
| In the past 1 year, have you had significant weight loss? [β] | | | | | |
| ○ Yes | 135 | 528 | 20 | 23.1 (19.8–26.8) | 92.6 (92.0–93.2) |
| ○ No | 449 | 6,650 | 6.3 | | |
| In the past 1 year, have you had a persistent or recurrent cough? | | | | | |
| ○ Yes | 81 | 321 | 19.4 | 13.9 (11.2–16.9) | 95.5 (95.0–96.1) |
| ○ No | 503 | 6,854 | 6.8 | | |
| In the past 1 year, have you had any symptoms or signs of an STI? * | | | | | |
| ○ Yes | 240 | 1,779 | 11.7 | 41.1 (37.1–45.2) | 75.2 (74.2–76.2)[^] |
| ○ No | 344 | 5,399 | 6.0 | | |
| Do you have a sexual partner who tested HIV positive in the last 2 years? | | | | | |
| ○ Yes | 399 | 3,411 | 10.4 | 68.3 (64.4–72.1) | 52.5 (51.3–53.6)[^] |
| ○ No | 185 | 3,767 | 4.6 | | |
| In the past 1 year, have you shared sharp objects with anyone? | | | | | |
| ○ Yes | 23 | 137 | 12.9 | 3.9 (2.5–5.9) | 98.1 (97.7–98.4) |
| ○ No | 561 | 7,041 | 6.9 | | |
| In the past 1 year, have you engaged in transactional sex? [α] | | | | | |
| ○ Yes | 100 | 664 | 12.9 | 17.1 (4.6–8.8) | 90.7 (90.1–91.4) |
| ○ No | 484 | 6,514 | 6.9 | | |
| In the past 2 years, have you been forced to have sex by anyone? | | | | | |
| ○ Yes | 35 | 236 | 12.4 | 6.5 (4.6–8.8) | 96.7 (96.3–97.1) |
| ○ No | 546 | 6,942 | 7.2 | | |
| In the past 1 year, have you had sex without a condom with an unusual partner, or the condom burst? | | | | | |
| ○ Yes | 877 | 621 | 12.1 | 14.9 (12.1–18.0) | 91.3 (90.7–92.0) |
| ○ No | 497 | 6,557 | 7.0 | | |

[#]The analysis was limited to clients who responded to all screening questions and had an HIV test result

[&]CI–confidence interval

[^]-Sensitivity >40%

[€]The dichotomous analysis combined all levels of risk (Mild, moderate and severe) against No risk

[α]Transactional sex included receiving money or goods in exchange for sex

[β]Significant weight loss was defined as unexplained weight loss >10%

[*]STI–Sexually Transmitted Infection

- Experiencing symptoms and signs of an STI within the previous 1 year was adjusted to 3 months and the syndromes specified in the SOP for guidance in quizzing participants

A standard operating procedure manual (SOP) was developed to guide health workers on the information to be elicited for each question. (**S3 Text**).

The following variables were collected: screening tool serial number, name of province, district and health facility, date, name of client, age, sex, previous HIV status, responses to the screening questions, acceptance for an HIV test, HIV test result. Pretesting of data collection tools was done at four Harare City Health Department clinics in Kuwadzana, Budiriro, Marlborough and Rujeko and adjusted for clarity of questions

## Analysis and statistics

Data were extracted from the paper-based screening forms and entered into excel, then read and analysed using EpiData Software Suite version 4.6 for, EpiData Association, Odense, Denmark) for descriptive and unadjusted analysis (**S4 Text**). The multivariable-adjusted analysis was done using STATA (version 12.1 STATA Corp., College Station, TX, USA).

Key analytic outputs were the client responses against the HIV test results to determine positivity yield, sensitivity and specificity, predictive value positive and negative (PVP & PVN), calculated reduction in testing volumes and the number needed to test (NNT). Clients who had previously tested HIV positive, missing responses on questions or HIV test results were excluded from the analysed data.

We explored combining different questions until the best output results of sensitivity were achieved, according to how participants had responded to these questions. The predetermined overall sensitivity for an acceptable screening tool was 90%. The criteria for screening in clients was determined by combining the testing interval and a positive response to any of the subsequent questions. Each scenario was then analysed to determine the same parameters measured at individual questions and combinations.

## Ethics approval and consent to participate

Approval to conduct this study was obtained from the Ministry of Health and Child Care head office, Joint Research Ethics Committee for the University of Zimbabwe Faculty of Medicine and Health Sciences and Parirenyatwa Group of Hospitals (JREC 280/2021) and the Medical Research Council of Zimbabwe (MRCZ/A/2783). Verbal informed consent was obtained from all participants before administering the tool and the study was conducted as part of routine service provision for HIV testing and adhering to stipulated standard guidelines.

## Results

### Evaluation findings

**Screening questions performance.** A total of 7 825 participants were recruited in 64 health facilities to respond to 13 selected screening questions. The overall positivity yield was 7.52% (584/7 825). Self-perception of risk scored the highest sensitivity of 86.8% (95% CI:83.8,89.5) with a specificity of 26.1% (95%CI:25.1,27.1). The question was analysed as a dichotomous variable after combining all levels of risk (Mild, moderate and severe) contrasted with clients who indicated no risk. Having a sexual partner who tested HIV positive within the last 2 years came second with a sensitivity of 68.3 (95%CI:64.4,72.1) followed closely a question on previous HIV test being at least 1 year ago at 63.4% sensitivity (95%CI:59.3,67.3). Having experienced poor health the within the previous 1 year came 4th, 42.1% (95%CI:38.1,46.2),

followed by having experienced symptoms and signs of an STI within the past year, 41.1% (95%CI:37.1,45.2). (**Table 1**).

Combining last tested >1 year with self-perception of risk, a partner who tested HIV positive, signs and symptoms of an STI and ill health in the past year produced a sensitivity of 95.7% (95%CI:93.1,98.4), reducing testing volumes by 14% but screening out 5% potential positive testers. This combination formed the draft screening tool that was validated.

## Validation findings

**Baseline characteristics.** We recruited a total of 2 140 participants for the validation phase. Females constituted 73.6% (n = 1 575) and 62.9% (n = 1,346) clients belonged to the age-group 20–34 years. (**Table 2**). The median age of clients was 27 years with an Interquartile range of 22–36 years. HIV testing yielded an overall 8% positivity (n = 171) whilst 0.7% (n = 14) inconclusive results and the rest obtained negative HIV test results. Previously, 2 clients (0.1%) had obtained a positive HIV test result.

**HIV positivity yield.** Of the 2116 clients that were screened and tested for HIV, 169 tested positive, translating to a positivity yield of 7.98%. (**Table 3**). Males recorded a higher positivity yield of 11.3% (n = 63) compared with their female counterparts at 6.7% (n = 105). Disaggregated by age group, the 35-49-year-olds recorded the highest positivity yield of 12.8% (n = 57), followed by the >/ = 50-year olds who recorded 11.4% (n = 9). A total of 14 (0.7%) of the test results were inconclusive whilst.

**Individual performance of screening questions.** *Previous HIV test period.* The majority of retests 35.2% (n = 752) were done within 3–6 months from the previous test whilst the

**Table 2. Clinical and demographic profile of the patients screened for HIV testing in Zimbabwe, 2021.** (N = 2140).

| Variable | Number | (%)* |
|---|---:|---|
| **Total** | **2140** | **(100)** |
| Age in years | | |
| ○ 16–19 | 267 | (12.5) |
| ○ 20–34 | 1346 | (62.9) |
| ○ 35–49 | 447 | (20.9) |
| ○ >/ = 50 | 80 | (3.7) |
| Median Age (IQR)^ | 27 | (22;35) |
| Gender | | |
| ○ Male | 564 | (26.4) |
| ○ Female | 1575 | (73.6) |
| ○ Not documented | 1 | (<0.1) |
| HIV test result | | |
| ○ Negative | 1952 | (91,3) |
| ○ Positive | 171 | (8.0) |
| ○ Inconclusive | 14 | (0.7) |
| Previous HIV Test Result# | | |
| ○ Negative | 2130 | (99.5) |
| ○ Positive | 2 | (0.1) |
| ○ Inconclusive | 8 | (0.4) |

*Column percentage

^IQR–Inter Quartile Range

#HIV test results obtained before the current visit

**Table 3. HIV test results by category and response to questions for clients screened in Zimbabwe, 2012–16.** (N = 2137) [#].

| Variable | HIV Test Results | | | | | |
|---|---:|---|---:|---|---:|---|
| | **Negative** | **(%)**[*] | **Positive** | **(%)**[*] | **Inconclusive** | **(%)**[*] |
| Total | 1952 | (91.3) | 171 | (8.0) | 14 | (0.7) |
| Age in years | | | | | | |
| ○ 16–19 | 251 | (94.7) | 14 | (5.3) | 0 | (<0.1) |
| ○ 20–34 | 1243 | (92.8) | 89 | (6.6) | 8 | (0.6) |
| ○ 35–49 | 385 | (86.5) | 57 | (12.8) | 3 | (0.7) |
| ○ >/ = 50 | 68 | (86.1) | 9 | (11.4) | 2 | (2.5) |
| Gender | | | | | | |
| ○ Female | 1461 | (93.0) | 105 | (6.7) | 5 | (0.3) |
| ○ Male | 486 | (87.3) | 63 | (11.3) | 8 | (1.4) |
| ○ Not documented | 0 | - | 1 | (100) | 0 | - |
| When was the last time you were tested for HIV? | | | | | | |
| ○ 0–3 months | 596 | (96.8) | 17 | (2.8) | 3 | (0.5) |
| ○ 3–12 months | 685 | (91.1) | 60 | (8.0) | 7 | (0.9) |
| ○ >12 months | 518 | (88.4) | 65 | (11.1) | 3 | (0.5) |
| ○ Never | 148 | (84.6) | 27 | (15.4) | 0 | - |
| Do you consider yourself to be at risk of HIV infection? | | | | | | |
| ○ No | 1373 | (93.6) | 85 | (5.8) | 9 | (0.6) |
| ○ Yes | 574 | (86.7) | 84 | (12.7) | 4 | (0.6) |
| Do you have a sexual partner who tested HIV positive in the last 2 years? | | | | | | |
| ○ No | 1815 | (92.7) | 132 | (6.7) | 11 | (0.6) |
| ○ Yes | 126 | (76.8) | 36 | (22.0) | 2 | (1.2) |
| Have you experienced poor health in the past 3 months? | | | | | | |
| ○ No | 1,786 | (93.4) | 116 | (6.1) | 11 | (0.6) |
| ○ Yes | 158 | (74.2) | 53 | (24.9) | 2 | (0.9) |
| Have you experienced any symptoms or signs of an STI? | | | | | | |
| ○ No | 1750 | (92.3) | 134 | (7.1) | 11 | (0.6) |
| ○ Yes | 195 | (84.1) | 35 | (15.1) | 2 | (0.8) |

[*] Row percentages

[#] The analysis was limited to clients who previously tested HIV negative or inconclusive

highest positivity yield, 15.4% (n = 27) was obtained from clients who had never tested for HIV. (**Table 3**). The probability of obtaining a positive test result was 54.4% (95% CI:46.6,62.1), with a specificity of 65.8% (95%CI:63.6,67.9) among the clients who had last tested more than a year ago, combined with clients who had never tested. (**Table 4**). The same category scored a predictive value positive of 12.1% (95%CI:9.9,14.7) and a predictive value negative of 94.3% (95%CI:93.0,95.5). Applying this category as a screening criterion reduced the testing volume by 64% and a number needed to test (NNT) of 8. (**Table 5**).

Among the clients who last tested for HIV less than 3 months, 2.8% (n = 17) tested positive indicating a sensitivity of 10% (95%CI:6.0,15.6) and specificity of 69.4% (95%CI:67.3,71.4). (**Tables 3 & 4**). The same category had a predictive value positive of 2.8% (95%CI:1.6,4.4) and a predictive value negative of 89.9% (95%CI:88.3,91.4). Applying this category as a screening criterion reduced the testing volume by 71% and require 36 tests to yield these attributes (NNT). (**Tables 4 & 5**).

*Self-perception of HIV risk*. A total of 662 clients (31.1%) responded yes to this question from which 84 (12.7%) tested HIV positive. (**Table 3**). The question was 49.7% sensitivity

**Table 4. Sensitivity and specificity for screening questions, Zimbabwe, 2021.** (N = 2116) [#].

| Variable | HIV Positive | TP+FP[*] | TN+FN[**] | Reduced tests (%) | Sensitivity % (95% CI) [&] | Specificity % (95% CI) |
|---|---|---|---|---|---|---|
| Total | 169 | | | | | |
| When was the last time you were tested for HIV? | | | | | | |
| ○ <3 months | 17 | 613 | 1503 | 71 | 10.0 (6.0–15.6) | 69.4 (67.3–71.4) |
| ○ 3-12months | 77 | 1358 | 758 | 36 | 45.6 (37.9–53.4) | 34.2 (32.1–36.4) |
| ○ >/ = 12months or never | 92 | 758 | 1358 | 64 | 54.4 (46.6–62.1) | 65.8 (63.6–67.9)[^] |
| Do you consider yourself to be at risk of HIV infection? | | | | | | |
| ○ Yes | 84 | 658 | 1458 | 69 | 49.7 (41.9–57.5) | 70.5 (68.4–72.5) |
| Do you have a sexual partner who tested HIV positive in the last 2 years? | | | | | | |
| ○ Yes | 36 | 162 | 1947 | 92 | 21.4 (15.5–28.4) | 93.5 (92.3–94.6) |
| Have you experienced poor health in the past 3 months? | | | | | | |
| ○ Yes | 53 | 211 | 1902 | 90 | 31.4 (24.5–38.9) | 91.9 (90.6–93.0) |
| Have you experienced any symptoms or signs of an STI?[***] | | | | | | |
| ○ Yes | 35 | 230 | 1884 | 89 | 20.7 (14.9–27.6) | 90.0 (88.6–91.3) |

[#]The analysis was limited to clients who responded to all screening questions and had an HIV test result

[&]CI–confidence interval

[*]TP+FP–True positives and False Positives

[**]TN+FN–True Negatives and False Negatives

[***]STI–Sexually Transmitted Infection

[^]Clients who last tested more than a year ago and those who had never tested were combined during the analysis

**Table 5. Predictive value positive and predictive value negative for HIV testing services screening questions, Zimbabwe, 2021.** (N = 2116) [#].

| Variable | HIV Positive | TP+ FP[*] | TN+ FN[**] | PVP[€] (95%CI) [&] | PVN[α] (95%CI) | NNT[β] |
|---|---|---|---|---|---|---|
| Total (n = 2,116) | 169 | | | | | 13 |
| When was the last time you were tested for HIV? | | | | | | |
| ○ <3 months | 17 | 613 | 1503 | 2.8 (1.6–4.4) | 89.9 (88.3–91.4) | 36 |
| ○ 3–12 months | 77 | 1358 | 758 | 5.7 (4.5–7.0) | 87.9 (85.3–90.1) | 18 |
| ○ >/ = 12 months or never | 92 | 758 | 1358 | 12.1 (9.9–14.7) | 94.3 (93.0–95.5) | 8 |
| Do you consider yourself to be at risk of HIV infection? | | | | | | |
| ○ Yes | 84 | 658 | 1458 | 12.8 (10.3–15.6) | 94.2 (92.8–95.3) | 8 |
| Do you have a sexual partner who tested HIV positive in the last 2 years? | | | | | | |
| ○ Yes | 36 | 162 | 1947 | 22.2 (16.1–29.4) | 93.2 (92.0–94.3) | 5 |
| Have you experienced poor health in the past 3 months? | | | | | | |
| ○ Yes | 53 | 211 | 1902 | 25.1 (19.4–31.5) | 93.9 (92.7–94.9) | 4 |
| Have you experienced any symptoms or signs of an STI?[***] | | | | | | |
| ○ Yes | 35 | 230 | 1884 | 15.2 (10.8–20.5) | 92.9 (91.6–94.0) | 7 |

[#]The analysis was limited to clients who responded to all screening questions and had an HIV test result

[€]Predictive Value Positive

[α]Predictive Value Negative

[&]CI–confidence interval

[*]TP+FP–True positives and False Positives

[**]TN+FN–True Negatives and False Negatives

[β]Number Needed to Test

[***]STI–Sexually Transmitted Infection

[^]Clients who last tested more than a year ago and those who had never tested were combined during the analysis

(95%CI:41.9,57.5) and 70.5% specific (95%CI:68.4,72.5), with a predictive value positive of 12.8% (95%CI:10.3,15.6) and predictive value negative of 94.2%. Applying this category as a screening criterion reduced testing volumes by 69% and require 8 tests to yield these attributes (NNT). **(Tables 4 & 5)**.

*A sexual partner who tested HIV positive within the last 2 years*. Of the 164 clients (7.7%) who responded yes to this question, 36 (22%) tested HIV positive. **(Table 3)**. The question was 21.4% sensitive (95%CI:15.5,28.4) and 93.5% specific (95%CI:92.3,94.6), with a predictive value positive of 22.2% (95%CI:16.1,29.4) and predictive value negative of 93.2% (95% CI:92.0,94.3). This question reduced the number of tests by 92% and the number needed to test was 5. **(Tables 4 & 5)**.

*Experienced poor health in the past 3 months*. Ill health included presumptive TB symptoms such as night sweats and productive cough for at least 2 weeks or unexplained weight loss of more than 10%. Among the 213 clients (10%) who responded yes to this question, 53 (24.9%) tested HIV positive. **(Table 3)**. The probability of obtaining a positive HIV diagnosis using this question was 31.4% (95%CI:24.5,38.9), and the specificity was 91.9% (95%CI:90.6,93.0), whilst a predictive value positive of 25.1%, (95%CI:19.4,31.5) and predictive value negative of 93.9% (95%CI:92.7,94.9) was achieved. Applying this category as a screening criterion reduced testing volumes by 90% and require 4 tests to yield these attributes (NNT). **(Tables 4 & 5)**.

*Experienced symptoms and signs of an STI*. The symptoms and signs measured by this question included urethral discharge, genital sores and other syndromic manifestations of STIs. A total of 2127 clients responded to this question of which 232 (10.9%) responded with a yes, resulting in a positivity yield of 15% (n = 35). **(Table 3)**. The probability of obtaining an HIV positive diagnosis using this question was 20.7% (95%CI:14.9,27.6) and 90% (95%CI:88.6,91.3) specific. A predictive value positive of 15.2% (95%CI:10.8,20.5) and predictive value negative of 92.9% (95%CI:91.6,94.0) was documented. Applying this question as a screening criterion reduced testing volumes by 89% and require 7 tests to yield these attributes (NNT). **(Tables 4 & 5)**.

**The combined performance of screening questions.** *Self-perceived risk*, *positive sexual partner*, *poor health in the past 3 months and having symptoms and signs of an STI*. Combining self-perceived risk with a positive sexual partner, poor health in the past 3 months and having symptoms and signs suggestive of an STI increased the probability of obtaining a positive HIV diagnosis to 69.2% (95%CI:61.7,76.1) and specificity of 61.6% (95%CI:59.4,63.8). **(Table 6)**. The predictive value positive for this combination was 13.5% (95%CI, 11.3–16.0) and a predictive value negative of 95.8% (95%CI:94.6,96.9). This combination reduced the testing volumes by 59% and the number needed to test was 7. **(Tables 6 & 7)**.

*Self-perceived risk*, *positive sexual partner*, *poor health*, *symptoms and signs of an STI and tested >/ = 3months*. Adding clients who last tested 3 months ago or longer raised the sensitivity to 94.1% (95%CI:89.4,97.1), reducing testing volumes by 18% and requiring 11 clients to be tested (NNT). **(Tables 6 & 7)**. This scenario was preferred, on account of its attributes on sensitivity and inclusivity on the period of the last test, but with a need to determine criteria for screening in and out.

*Self-perceived risk*, *positive sexual partner*, *poor health*, *symptoms and signs of an STI and last tested >12 months or never*. Changing the last tested period to more than 12 months or never reduced the sensitivity to 84% (95%CI:77.6,89.2), reducing the testing volumes by 39% and requiring 9 clients to be tested (NNT). **(Tables 6 & 7)**.

**Recommended criteria for screening in and out.** Considering the results of the second scenario of combined questions above (Self-perceived risk, positive sexual partner, poor health, symptoms and signs of an STI and tested >/ = 3months), 4-pronged criteria for screening in

**Table 6. Sensitivity and specificity for combined HIV testing services screening questions, Zimbabwe, 2021.** (N = 2116).

| Variable | HIV Positive | TP+FP* | TN+FN** | Reduced tests (%) | Sensitivity % (95% CI) [&] | Specificity % (95% CI) |
|---|---|---|---|---|---|---|
| Total | **169** | | | | | |
| Combined Questions (Self-perceived risk, Positive sexual partner, Poor Health, STI)*** | | | | | | |
| ○ Yes | 117 | 864 | 1252 | 59 | 69.2 (61.7–76.1) | 61.6 (59.4–63.8) |
| Combined Questions (Self-perceived risk, Positive sexual partner, Poor Health, STI) and tested >/ = 3 months | | | | | | |
| ○ Yes | 159 | 1727 | 389 | 18 | 94.1 (89.4–97.1) | 19.5 (17.7–21.3) |
| Combined Questions (Self-perceived risk, Positive sexual partner, Poor Health, STI) and tested >12 months or never[^] | | | | | | |
| ○ Yes | 142 | 1286 | 830 | 39 | 84.0 (77.6–89.2) | 41.2 (39.0–43.5) |

[&]CI–confidence interval

*TP+FP–True positives and False Positives

**TN+FN–True Negatives and False Negatives

***STI–Sexually Transmitted Infection

[^]Clients who last tested more than a year ago and those who had never tested were combined during the analysis

and out was devised which require a previous testing period of at least 3 months, combined with a yes response to any of the remaining 4 questions. **(Tables 8 & 9)**.

*Last tested >/ = 3months and self-perceived risk.* This criterion obtained a sensitivity of 93.5% (95%CI:88.7,96.7), predictive value positive of 9.4% (95%CI:8.0,10.9), reducing testing volumes by 20% and requiring 11 tests (NNT) to yield these attributes. **(Tables 8 & 9).**

*Last tested >/ = 3months and sexual partner positive.* This criterion scored a sensitivity of 89.9% (95%CI:84.4,94.0), predictive value positive of 9.8% (95%CI:8.4,11.4), reducing testing volumes by 27% and requiring 10 tests (NNT) to yield these attributes. **(Tables 8 & 9)**.

*Last tested >/ = 3months and poor health.* The probability of obtaining a positive HIV diagnosis by screening clients using a previous test period of at least 3 months combined with self-perception of risk was 91.1%, (95%CI:85.8,94.9), predictive value positive of 10% (95% CI:8.5,11.6), reducing testing volumes by 27% and requiring 10 tests (NNT) to obtain these attributes. **(Tables 8 & 9)**.

**Table 7. Predictive value positive and predictive value negative for combined HIV testing services table screening questions, Zimbabwe, 2021.** (N = 2116).

| Variable | HIV Positive | TP+ FP* | TN+ FN** | PVP[€] (95%CI)[&] | PVN[α] (95%CI) | NNT[β] |
|---|---|---|---|---|---|---|
| Total | 169 | | | | | 13 |
| Combined Questions (Self-perceived risk, Positive sexual partner, Poor Health, STI***) | | | | | | |
| ○ Yes | 117 | 864 | 1252 | 13.5 (11.3–16.0) | 95.8 (94.6–96.9) | 7 |
| Combined Questions (Self-perceived risk, Positive sexual partner, Poor Health, STI) and tested >/ = 3 months | | | | | | |
| ○ Yes | 159 | 1727 | 389 | 9.2 (7.9–10.7) | 97.4 (95.3–98.8) | 11 |
| Combined Questions (Self-perceived risk, Positive sexual partner, Poor Health, STI) and tested >12 months or never[^] | | | | | | |
| ○ Yes | 142 | 1286 | 830 | 11.0 (9.4–12.9) | 96.7 (95.3–97.8) | 9 |

[€]Predictive Value Positive

[α]Predictive Value Negative

[&]CI–confidence interval

*TP+FP–True positives and False Positives

**TN+FN–True Negatives and False Negatives

[β]Number Needed to Test

***STI–Sexually Transmitted Infection

[^]Clients who last tested more than a year ago and those who had never tested were combined during the analysis

**Table 8. Sensitivity and specificity for inclusion criteria combinations for HIV testing services screening questions, Zimbabwe, 2021.** (N = 2116).

| Variable | HIV Positive | TP+ FP* | TN+FN** | Reduced tests (%) | Sensitivity % (95% CI)[&] | Specificity % (95% CI) |
|---|---|---|---|---|---|---|
| Total | 169 | | | | | |
| Last tested >/ = 3 months and Risk | | | | | | |
| ○ Yes | 158 | 1687 | 429 | 20 | 93.5 (88.7–96.7) | 21.5 (19.7–23.4) |
| Last tested >/ = 3 months and Sexual partner positive | | | | | | |
| ○ Yes | 152 | 1553 | 563 | 27 | 89.9 (84.4–94.0) | 28.0 (26.1–30.1) |
| Last tested >/ = 3 months and Poor health | | | | | | |
| ○ Yes | 154 | 1540 | 576 | 27 | 91.1 (85.8–94.9) | 28.8 (26.8–30.9) |
| Last tested >/ = 3 months and STI*** symptoms | | | | | | |
| ○ Yes | 155 | 1561 | 555 | 26 | 91.7 (86.5–95.4) | 27.8 (25.8–29.8) |

[&]CI–confidence interval

*TP+FP–True positives and False Positives

**TN+FN–True Negatives and False Negatives

***STI–Sexually Transmitted Infections

*Last tested >3/ = months and STI symptoms.* The probability of obtaining a positive HIV diagnosis by screening clients using a previous test done at least 3 months previously, combined with signs and symptoms of an STI was 91.7% (95%CI:86.5,95.4), predictive value positive of 9.9% (95%CI:8.5,11.5), reducing testing volumes by 26% and requiring 10 tests (NNT) to yield these attributes. **(Tables 8 & 9)**.

## Discussion

This study analyzed the individual performance of screening questions before analyzing suggested combinations and then proposing a screening in and out criteria that resulted in a highly sensitive (94%) adult HTS screening tool (**S4 Text**). The tool reduced HIV testing

**Table 9. Predictive value positive and predictive value negative for inclusion criteria combinations for HIV testing services screening questions, Zimbabwe, 2021.** (N = 2116).

| Variable | HIV Positive | TP+FP* | TN+FN** | PVP[€] (95%CI)[&] | PVN[α] (95%CI) | NNT[β] |
|---|---|---|---|---|---|---|
| Total | 169 | | | | | 11 |
| Last tested >/ = 3 months and Self-perceived risk | | | | | | |
| ○ Yes | 158 | 1687 | 429 | 9.4 (8.0–10.9) | 97.4 (95.5–98.7) | 11 |
| Last tested >/ = 3 months and Sexual partner positive | | | | | | |
| ○ Yes | 152 | 1553 | 563 | 9.8 (8.4–11.4) | 97.0 (95.2–98.2) | 10 |
| Last tested >/ = 3 months and Poor health | | | | | | |
| ○ Yes | 154 | 1540 | 576 | 10.0 (8.5–11.6) | 97.4 (95.7–98.5) | 10 |
| Last tested >/ = 3 months and STI*** symptoms | | | | | | |
| ○ Yes | 155 | 1561 | 555 | 9.9 (8.5–11.5) | 97.5 (95.8–98.6) | 10 |

[€]Predictive Value Positive

[α]Predictive Value Negative

[&]CI–confidence interval

*TP+FP–True positives and False Positives

**TN+FN–True Negatives and False Negatives

[β]Number Needed to Test

***STI–Sexually Transmitted Infection

volumes by 18%, required 11 tests as the number needed to test. However, the tool also screened out 6% of potential positives. This finding is at variance with Moucheraud et al (2021) who found no "clear" advantages of screening tools in Malawi [17].

## Strengths

The consistent availability of the HIV testing services at health care facilities facilitated the testing of clients following the screening, which was critical for the analysis to match responses to questions with the HIV test result. Including all clients that visited HIV testing points at the health care facilities in the study minimized sampling bias and resulted in a large sample size to generalize the findings. The study was done using routinely collected programmatic data which was representative of the reality on the ground.

## Limitations

The analysis was restricted to clients who responded to all questions and also accepted to be tested and further obtained a valid HIV test result. This resulted in a total of 89 participants being excluded from the analysis, at various stages of the data cleaning process. The potential contribution of these participants was therefore missed. Secondly, where clients responded yes to more than one question, it was not possible to ascertain the contribution of the individual questions to HIV test outcome.

## Interpretation of key findings

This study provided important insights into the performance of the HIV Testing Screening for eligibility for testing in Zimbabwe.

First, HIV positivity yield remained comparable during the time of the study and before the evaluation, 7.52% during evaluation and 7.98% during validation **(Tables 1 & 2)**, against 7.83% (Programmatic data for the same sites for March 2021). This was an observation of concern. Considering that a screening tool was already being utilized to determine eligibility for testing, with clients being screened in and others being screened out, it was expected that during the evaluation, the positivity yield would have declined since all clients were being tested, regardless of the screening outcome. This finding influenced us to hypothesize that either the screening tool was not being routinely utilized to determine eligibility for testing, or the tool was being used and yet not effective to correctly screen in and out, which is also supported by the fact that the tool had not been evaluated.

Second, the individual performance of the screening questions ranged from a lowest of 10% for clients who previously tested within 3 months to a highest of 90% among clients who experienced ill health within the previous 3 months **(Table 2)**. The five questions made it into the validation process following the evaluation phase which eliminated the questions which had not performed well. This finding indicated the importance of developing a screening in criteria that combine more than 1 question's response to enhance performance. Further, this also indicated the lack of value in testing clients at an interval of fewer than 3 months. The high probability of obtaining a negative HIV test result within 3 months of a previous test is well documented in the literature, in the context of the window period of seroconversion [18–20]. This finding became an important contributor to the suggested screening in combinations to start at a testing interval of at least 3 months.

Third, exploring different combinations of questions enabled us to exhaust the potential of the screening tool. Though the performance met the set threshold for sensitivity, it was observed that these attributes were overall and a screening in and out criteria was needed. The

likelihood of a client responding yes to all the questions was less likely and therefore, effective implementation of the tool required considering a criterion for screening in and out.

Fourth, the screening in criteria was obtained by considering the overall performance of a combination of questions and then going further to analyze the attributes of each screening in criteria, which combined a testing interval of at least 3 months and a yes response to any of the subsequent 4 questions. The screening criteria requires clear guidance to be provided to the health worker to ensure its adhered to. Therefore a standard operating procedure (SOP) was developed to aid this process.

Finally, the obtained attributes of the screening tool (94.1% sensitivity, 18% reduction in testing volumes and NNT of 11) met and surpassed the set criteria at the onset, and presents a favourable route for effective use of finite resources in a severely constrained environment. This finding was inconsistent with Antelman et al (2021) in Tanzania who observed a high NNT despite changes in yield [21]. However, the leakage of 6% for clients who will be wrongly screened out presents a challenge on innovations that can be employed to bring them to care. The first option is a frequent re-screening schedule (3 months) to timely identify them. The utility of HIVST was also explored with the challenge being the fact that HIVST is distributed in a targeted manner in the country. Offering HIVST to clients who get screened out but insist on an HIV test was explored and found feasible.

## Implications for policy and practice

The use of screening tools to assess eligibility for testing is essential in a resource-constrained setting with a culture of repeated tests within short time intervals (<3months) which does not account for risk profiling. Screening for eligibility of an HIV test mitigates resource constraints through testing clients with a high probability of turning out positive, according to their risk profile. This minimizes "unnecessary testing", where a negative diagnosis is almost predictable as seen in clients whose testing intervals were less than 3 months (sensitivity; 10% and NNT of 36). The developed screening tool proposes a shift in how the screening process for HIV testing is conducted. This is a shift from determining eligibility for testing through a yes response from any of the 7 previous questions whose properties were unknown, to utilizing defined criteria for determining eligibility for HIV testing using a 5-question screening tool whose properties are now documented.

For a screening tool to be effective, the document must be adopted as a national document, and be applied at all HIV testing entry points to ensure that its attributes are realized at the national level. The utilisation of the tool needs to be fostered among health workers, with strict adherence to the SOP for screening criteria. Sensitization of health workers on the importance and effective utilisation of the screening tool, strictly applying the SOP guidance on the screening in criteria, is imperative.

The HTS program in Zimbabwe should consider innovative ways to capture clients who are screened out and would have otherwise tested HIV positive. Assuming that they will be captured through other platforms such as Index testing and re-screening at subsequent visits may not be adequate for timely identification of people living with HIV, considering the risk of onward transmission in the community. Providing HIVST to clients who insist on testing after being screened out was explored as a feasible option, though it would be an adjustment to the current targeted distribution of HIVST kits protocol.

There is a need to migrate to EHR and have the tool being part of the client flow and guiding health workers on screening in criteria and timelines for re-assessment among those screened out at the initial visit. Provision of a printed version of the screening tool at all HTS entry points translated to vernacular is essential for sites that are yet to migrate to eHR.

### Future research

First, a qualitative study to explore the health worker perspectives and utilization of HTS screening tools is required. Second, once the tool is now routinely utilized at health care facilities, operational research may be considered to evaluate how the tool will be performing to observe if the findings of this study will translate to the application at the national level.

## Conclusion

We successfully developed an adult HTS screening tool through an evaluation and validation process. The attributes of the output (94.1% sensitivity, 18% reduction in testing volume and 11 NNT) were accepted to improve testing efficiency whilst reducing testing volumes. This tool provides a scientific way of determining eligibility for testing which is critical in targeting HIV testing to increase positivity yield and expedite meeting espoused HIV testing milestones. Innovations to accommodate 6% of potential positive testers who are screened out were explored, including using HIV self-test who insist on testing after being screened out.

## Supporting information

**S1 Text. Screening tool evaluation questions.**
(DOCX)

**S2 Text. Questions for screening tool validation.**
(DOCX)

**S3 Text. HTS screening tool SOP.**
(DOCX)

**S4 Text. Finalized screening tool.**
(DOCX)

**S1 Data. Zimbabwe adult screening tool dataset.**
(XLSX)

## Acknowledgments

I acknowledge several individuals and institutions that made this study a success. Special gratitude goes to my academic supervisors, Professor M. Tshimanga, Dr J. Chirenda and Dr K. Takarinda, The Director for AIDS & TB Unit, Dr O. Mugurungi and the entire HTS team for their support and prodding during this study.

## Author Contributions

**Conceptualization:** Hamufare Dumisani Mugauri, Joconiah Chirenda, Kudakwashe Takarinda, Owen Mugurungi, Blessing Mushangwe, Mufuta Tshimanga.

**Data curation:** Hamufare Dumisani Mugauri, Joconiah Chirenda, Kudakwashe Takarinda, Ishmael Chikondowa, Patrick Mantiziba, Blessing Mushangwe.

**Formal analysis:** Hamufare Dumisani Mugauri, Joconiah Chirenda, Kudakwashe Takarinda, Ishmael Chikondowa, Patrick Mantiziba, Blessing Mushangwe.

**Funding acquisition:** Hamufare Dumisani Mugauri, Owen Mugurungi, Getrude Ncube, Patrick Mantiziba, Blessing Mushangwe.

**Investigation:** Hamufare Dumisani Mugauri, Joconiah Chirenda, Getrude Ncube, Ishmael Chikondowa, Patrick Mantiziba, Blessing Mushangwe, Mufuta Tshimanga.

**Methodology:** Hamufare Dumisani Mugauri, Joconiah Chirenda, Kudakwashe Takarinda.

**Project administration:** Hamufare Dumisani Mugauri, Owen Mugurungi.

**Resources:** Hamufare Dumisani Mugauri, Getrude Ncube, Patrick Mantiziba.

**Software:** Kudakwashe Takarinda, Ishmael Chikondowa, Patrick Mantiziba.

**Supervision:** Joconiah Chirenda, Kudakwashe Takarinda, Owen Mugurungi, Mufuta Tshimanga.

**Validation:** Hamufare Dumisani Mugauri, Joconiah Chirenda, Kudakwashe Takarinda, Ishmael Chikondowa, Patrick Mantiziba, Mufuta Tshimanga.

**Visualization:** Hamufare Dumisani Mugauri, Joconiah Chirenda, Kudakwashe Takarinda, Owen Mugurungi, Getrude Ncube, Ishmael Chikondowa, Patrick Mantiziba, Blessing Mushangwe, Mufuta Tshimanga.

**Writing – original draft:** Hamufare Dumisani Mugauri.

**Writing – review & editing:** Hamufare Dumisani Mugauri, Mufuta Tshimanga.

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
