## [Decision Letter · Decision Letter 0]

16 Mar 2022

PGPH-D-22-00259

Optimizing the Adult HIV Testing Services Screening tool to predict positivity yield in Zimbabwe, 2021

Dear Mugauri,

Thank you for submitting your manuscript to PLOS Global Public Health. After careful consideration, we feel that it has merit but does not fully meet PLOS Global Public Health’s publication criteria as it currently stands. Therefore, we invite you to submit a revised version of the manuscript that addresses the points raised during the review process.

We look forward to receiving your revised manuscript.

Kind regards,

Collins Otieno Asweto, PhD

Academic Editor

Journal Requirements:

1. Please ensure that the Title in your manuscript file and the Title provided in your online submission form are the same.

2. Please amend your detailed Financial Disclosure statement. This is published with the article, therefore should be completed in full sentences and contain the exact wording you wish to be published.

ii). State the initials, alongside each funding source, of each author to receive each grant.

iii). State what role the funders took in the study. If the funders had no role in your study, please state: “The funders had no role in study design, data collection and analysis, decision to publish, or preparation of the manuscript.”

iv). If any authors received a salary from any of your funders, please state which authors and which funders.

3. Please ensure that the funders and grant numbers match between the Financial Disclosure field and the Funding Information tab in your submission form. 

4. We have noticed that you have uploaded supporting information but you have not included a list of legends.  Please add a full list of legends for all supporting information files (including figures, table and data files) after the references list. 

Reviewers' comments:

Reviewer's Responses to Questions

**Comments to the Author**

1. Does this manuscript meet PLOS Global Public Health’s publication criteria? Is the manuscript technically sound, and do the data support the conclusions? The manuscript must describe methodologically and ethically rigorous research with conclusions that are appropriately drawn based on the data presented.

Reviewer #1: Yes

Reviewer #2: Partly

2. Has the statistical analysis been performed appropriately and rigorously?

Reviewer #1: Yes

Reviewer #2: Yes

3. Have the authors made all data underlying the findings in their manuscript fully available (please refer to the Data Availability Statement at the start of the manuscript PDF file)?

Reviewer #1: Yes

Reviewer #2: Yes

4. Is the manuscript presented in an intelligible fashion and written in standard English?

Reviewer #1: Yes

Reviewer #2: No

5. Review Comments to the Author

Reviewer #1: Testing is the gateway to care for people with HIV. Scientifically-proven, cost effective and scalable interventions are recommended by the CDC and WHO as a way to improve outcomes in HIV prevention especially in regions where HIV prevalence is high. The screening tool described in this study meets this threshold.

While the study uses a cross sectional survey, having a comparison group would have provided a stronger link between this tool and the positivity yield and whether the change observed would be directly attributed to the use of this tool. The change described in the conclusion of this study did not take place with a comparison group who did not receive the same intervention. As it is, we can not confidently conclude that the change we see is due to the screening tool alone. Therefore, the results we have here may hold but with a recommendation for future research to include a comparison group. Robustness of the findings was demonstrated by performing sensitivity analysis and this is a strength for the study.

Overall, the study objective is reflected in the summarized findings and the authors report a balanced record of the analyzed results.

Reviewer #2: This is a very interesting and important piece of work that will help many countries to adapt and tailor the screening tool for HIV. I would like to congratulate the authors on this work.

However, there are major issues with the way this paper has been written and designed:

1. The authors need to define upfront what the tools consisted of. After many paragraphs in the intro, it is said that it was a questionnaire. The tool should be defined and briefly described giving the overall view of the screening tool

2. The authors discuss that the tool was adapted and many other variables were added and sub categories were made, but this needs to be again described briefly much before the analysis subsection as the actual work is not clear before the analysis section and this can get difficult for the reader to understand.

3. in line 189 it is mentions, "Various combinations were tested....", this is again getting complicated as it is not clear what combinations are the authors talking about? The questions in the tool or the categories in the answers? Or the addition/deletion of questions in the tool to attain high sensitivity?

4. The discussion needs to succinctly mention overall changes that were made and how it impacted.

5. The discussion also needs to discuss how it increased positivity rates- were they more close to the actual HIV incidence in these regions? As the regions were randomly selected, the %positivity can be compared to these regions selected rather than the national incidence. This will help understand the impact of this modified screening tool better.

It is very difficult to understand the overall message that the authors are trying to tell. It is only later in the paper that the reader can understand what the actual message is.

Such an important piece of work needs more finesse in writing so that the message does not gets diluted.

6. PLOS authors have the option to publish the peer review history of their article (what does this mean?). If published, this will include your full peer review and any attached files.

**Do you want your identity to be public for this peer review?** For information about this choice, including consent withdrawal, please see our Privacy Policy.

Reviewer #1: **Yes: **Maurine Ng'oda

Reviewer #2: No

---

## [Decision Letter · Decision Letter 1]

18 May 2022

Optimising the Adult HIV Testing Services Screening tool to predict positivity yield in Zimbabwe, 2021

PGPH-D-22-00259R1

Dear Mugauri,

We are pleased to inform you that your manuscript 'Optimising the Adult HIV Testing Services Screening tool to predict positivity yield in Zimbabwe, 2021' has been provisionally accepted for publication in PLOS Global Public Health.

Best regards,

Collins Otieno Asweto, PhD

Academic Editor

Reviewer Comments (if any, and for reference):

Reviewer's Responses to Questions

**Comments to the Author**

1. If the authors have adequately addressed your comments raised in a previous round of review and you feel that this manuscript is now acceptable for publication, you may indicate that here to bypass the “Comments to the Author” section, enter your conflict of interest statement in the “Confidential to Editor” section, and submit your "Accept" recommendation.

Reviewer #1: All comments have been addressed

Reviewer #2: All comments have been addressed

2. Does this manuscript meet PLOS Global Public Health’s publication criteria? Is the manuscript technically sound, and do the data support the conclusions? The manuscript must describe methodologically and ethically rigorous research with conclusions that are appropriately drawn based on the data presented.

Reviewer #1: Yes

Reviewer #2: Yes

3. Has the statistical analysis been performed appropriately and rigorously?

Reviewer #1: Yes

Reviewer #2: Yes

4. Have the authors made all data underlying the findings in their manuscript fully available (please refer to the Data Availability Statement at the start of the manuscript PDF file)?

Reviewer #1: Yes

Reviewer #2: Yes

5. Is the manuscript presented in an intelligible fashion and written in standard English?

Reviewer #1: Yes

Reviewer #2: Yes

6. Review Comments to the Author

Reviewer #1: This is an interesting study and the authors have collected a unique dataset using an acceptable approach. The research question was clearly stated and generally, the manuscript is written well and correctly structured. Method of analysis was explained and appeared appropriate. Overall, the generated evidence validates the relevance and use of screening tools to assess eligibility for testing in a resource constrained settings in order to determine eligibility for testing which is critical in targeting HIV testing to increase positivity yield and expedite meeting espoused HIV testing milestones.

Reviewer #2: (No Response)

7. PLOS authors have the option to publish the peer review history of their article (what does this mean?). If published, this will include your full peer review and any attached files.

**Do you want your identity to be public for this peer review?** For information about this choice, including consent withdrawal, please see our Privacy Policy.

Reviewer #1: **Yes: **Maurine Ng'oda

Reviewer #2: No
